# Effects of Global Climate Warming on the Biological Characteristics of *Spodoptera frugiperda* (J.E. Smith) (Lepidoptera: Noctuidae)

**DOI:** 10.3390/insects15090689

**Published:** 2024-09-12

**Authors:** Chun Fu, Zhiqian Liu, Danping Xu, Yaqin Peng, Biyu Liu, Zhihang Zhuo

**Affiliations:** 1Key Laboratory of Sichuan Province for Bamboo Pests Control and Resource Development, Leshan Normal University, Leshan 614000, China; fuchun421@aliyun.com; 2College of Environmental Science and Engineering, China West Normal University, Nanchong 637002, China; qnhtvxhp319123@foxmail.com (Z.L.); xudanping@cwnu.edu.cn (D.X.); pengyaqin2023@foxmail.com (Y.P.); biyuliuql@foxmail.com (B.L.)

**Keywords:** *Spodoptera frugiperda*, climate change, integrated pest management, invasive insects, meta-analysis, subsistence analysis

## Abstract

**Simple Summary:**

*Spodoptera frugiperda* is an important economic pest that has recently invaded Africa and Asia. This study systematically assessed its life history traits under varying temperatures, photoperiods, and humidity levels. The results show that its physiological activity is highest at 32 °C, with reduced durations of the developmental stages, increased oviposition quantity and period in females, and a shorter pupal stage, which extends the adult lifespan. These findings help predict population dynamics and inform management strategies.

**Abstract:**

*Spodoptera frugiperda* (J.E. Smith) is a significant economic pest that has recently invaded Africa and Asia. However, much of the information regarding its ecological capabilities in these newly invaded environments remains largely unknown. In this study, the life history traits of the fall armyworm under conditions of increased temperature, different photoperiods, and varying humidity levels were systematically evaluated. Among 43 studies, a total of 20 studies were included in the analysis by passing the screening criteria, and random-effects meta-analysis, fixed-effects meta-analysis, and meta-regression were conducted. It has been found that with the increase in temperature above 20 °C, various physiological indicators of the fall armyworm are significantly enhanced. When the temperature reaches 32 °C, the physiological activities of *S. frugiperda* are at their highest point. As the temperature increases, the duration of each developmental stage of the fall armyworm decreases significantly, accompanied by an increase in oviposition quantity and period in females. Additionally, the pupal development time is shortened, which leads to an increase in the lifespan of the adult moth. Using temperature and relative humidity as environmental variables, the optimal survival conditions for each insect state of the fall armyworm were calculated. These findings can assist in predicting the population dynamics of the fall armyworm and in formulating appropriate and practical management strategies.

## 1. Introduction

Global climate change may significantly impact the geographic distribution of biodiversity, alter biodiversity, and affect interactions between species and ecosystems [1]. Due to their high reproductive capacity and abundance, invasive species are often better equipped to cope with climate change than native species [2,3]. As temperatures rise, temperature-driven changes are believed to alter the stability of ecosystems and maintain the generation cycles of diverse insects [4]. Therefore, rising temperatures may expand the global distribution of invasive crop pest species, trigger epidemics, and subsequently increase damage to crops [5,6,7,8]. The establishment and spread of invasive pests in new environments are typically attributed to the lack of natural enemies. However, the absence of co-evolved host plants in invaded areas also plays a significant role in facilitating pest invasions [9].

*Spodoptera frugiperda* (J.E. Smith) (Lepidoptera: Noctuidae) is one of the most destructive crop pests in the Americas. Its larvae can attack over 353 species of hosts from 76 families, primarily targeting plants in the Poaceae family (106 taxa) [10]. *S. frugiperda* has the ability of long-distance flight and has been confirmed as an invasive pest in Africa (47 countries), Asia (18 countries), and the continent of Australia as of February 2020 [11,12]. Due to the absence of diapause and its short generation time, populations of *S. frugiperda* may occur throughout the year in non-native countries where temperatures and host plants are suitable [13].

The fall armyworm is a highly polyphagous herbivorous insect, represented by two sympatric host plant strains: one primarily feeds on rice and various grasses, known as the rice strain, while the other mainly consumes maize, sorghum, cotton, and sugarcane, referred to as the maize strain [14,15]. The fall armyworm feeds on plants belonging to the Poaceae family, including sorghum, maize, and millet [16]. Sorghum is an important drought-tolerant crop in China, used for food, feed, and energy. In June 2019, an outbreak of fall armyworm infestations was observed in sorghum fields [17]. In 2019, *S. frugiperda* was found to be damaging Job’s tears (Coix lacryma-jobi) in Qichun County, Hubei Province, and Xingyi City, Guizhou Province, China. Job’s tears is a high-quality grain that is both edible and medicinal in China [18], accounting for over 70% of the global market. Therefore, it is necessary to explore the adaptability of *S. frugiperda* populations in invaded areas to abiotic stressors, such as the temperature, humidity, and photoperiod.

With the rapid changes in Earth’s climate, the ability of invasive species to regulate their performance is crucial for adapting to new invasion environments [19]. Temperature is the central regulatory abiotic variable controlling the development, reproductive capacity, and range of invasive species [20]. The developmental rate of invasive insects increases with rising temperatures [21]. Due the lack of diapause and suitable host plants, *S. frugiperda* could not previously overwinter in northern China and North America. However, high temperatures accelerate the growth of *S. frugiperda* and shorten their lifecycle, leading to outbreaks of *S. frugiperda* [22,23,24,25]. Therefore, it is necessary to evaluate the impact of temperature changes on the performance of *S. frugiperda*.

Although several studies have reported on the influence of temperature on fall armyworms, there has been no quantitative, comprehensive assessment of the impact of temperature changes on various physiological indicators of *S. frugiperda* to date. In this study, a meta-analysis was conducted to comprehensively evaluate the sensitivity of various physiological indicators of the fall armyworm to temperature changes under different temperature conditions.

## 2. Materials and Methods

### 2.1. Literature Search

In this study, we conducted targeted literature searches using CNKI, Web of Science, and Google Scholar, with “Temperature” and “*Spodoptera frugiperda* (J.E. Smith)” as the main keywords. By screening abstracts and full texts, 43 studies on the effects of temperature on fall armyworm were identified. Further screening of the full texts of these 43 studies was conducted based on included criteria, resulting in the final selection of 20 studies to be included in the research.

### 2.2. Inclusion Criteria

Studies that met the following criteria were included: (1) The research reports included the effects of temperature variations on the developmental period, lifespan, life cycle, oviposition period, fecundity, and eclosion rate of *S. frugiperda*. (2) In addition to temperature gradients, the research reports also contained data on relative humidity, photoperiod, and other variables. (3) The research reports contained sample sizes, means, standard errors, or standard deviations of the experimental data.

### 2.3. Data Retrieval

According to the inclusion criteria, a total of 605 datasets were collected. For each study, data were collected on the developmental period, lifespan, life cycle, oviposition period, fecundity, eclosion rate of *S. frugiperda*, and several regulating factors by observing changes in various physiological indicators. Each study’s control and treatment levels were defined based on temperature gradients, with the control level set as the lowest temperature condition. All temperature gradients except the lowest temperature were considered treatment groups. We investigated the changes in various physiological indicators of *S. frugiperda* under conditions of temperature increase. Using low temperature as the control and other temperature gradients as treatments, data were extracted to ensure that the collected data included the corresponding standard deviation (SD). If the data in the research report contained a standard error (SE), it needed to be converted using a formula, typically by dividing the standard error by the square root of the sample size to calculate the standard deviation [26]. The calculation method was as shown in Formula (1).
(1)$$\mathrm{SD}=\mathrm{SE}\sqrt{n}$$

In Formula (1), ‘n’ indeed represents the sample size.

### 2.4. Data Analysis

The log response ratio was chosen as the measure of effect size, which is represented by the standardized mean difference adjusted for positive skewness [27] (the original mean difference divided by the pooled standard deviation of the two groups). The log–response ratio is widely applied in ecology, enabling comparability of results with other studies. The effect size for each observation was computed as the natural logarithm transformed (*ln*) response ratio (*RR*), as in Formula (2):(2)$$RR=ln(\frac{X_{t}}{X_{c}})=ln(X_{t})-ln(X_{c})$$

In Formula (2), *X_t_* and *X_c_* represent the mean values of the developmental periods of *S. frugiperda* in the experimental group and control group, respectively. The variance (v) (Formula (3)) of the RR is calculated as follows:(3)$$v=\frac{s_{t}^{2}}{n_{t}{X_{t}}^{2}}+\frac{s_{c}^{2}}{n_{c}{X_{c}}^{2}}$$

In the Formula (3), *n_t_* and *n_c_* represent the sample sizes of the experimental and control groups, respectively. Similarly, *s_t_* and *s_c_* represent their respective standard deviations (SD). When SD or SE is not provided in the study, one tenth of the mean value is specified as the standard deviation. The weighted average response (*RR*^+^) is obtained by weighting the response ratio of each independent study (Formula (4)):(4)$$RR^{+}=\frac{\sum_{i=1}^{m}w_{i}(RR_{i})}{\sum_{i=1}^{m}w_{i}}$$

“*m*” is the number of comparisons in the group, *w_i_* is the weighting factor for the i-th experiment in the group, and *w*_i_ is calculated as follows (Formula (5)):(5)$$w_{i}=\frac{1}{v_{i}}$$

The standard error *s*(*RR*^+^) and 95% confidence interval (CI) are calculated as Formulas (6) and (7), respectively.
(6)$$s(RR^{+})=\sqrt{\frac{1}{\sum_{i=1}^{m}w_{i}}}$$
(7)$$95\%\mathrm{CI}=lnRR^{+}\pm 1.96s(lnRR^{+})$$

*Q_t_* is used to determine whether to introduce explanatory variables into the calculation results, according to the following (Formula (8)):(8)$$Q_{t}=Q_{m}+Q_{e}$$

In Formula (8), *Q_m_* represents the heterogeneity caused by a known factor, where a higher value indicates a greater influence of the explanatory variable on the effect size. On the other hand, *Q_e_* represents the unexplained residual heterogeneity. When the total heterogeneity *Q_t_* is high, it indicates that data points deviate significantly from the mean, possibly due to other factors causing such deviations. If the data are homogeneous, *Q_t_* should follow a chi-square distribution with k-1 degrees of freedom, and in this case, there is no need to introduce explanatory variables.

### 2.5. Meta-Analysis

In this study, the “rma.mv” function from the R package “metafor” in version 4.3 was employed for the following steps [28]. Firstly, the relative risk (*RR*^+^) was calculated using a random-effects model, and the variance between cases was estimated using restricted maximum likelihood (REML) [29]. Explanatory variables were introduced based on the value of I^2^. Subsequently, a random-effects model was used to compute the overall average effect size for all treatment group temperatures. Finally, all statistical tests were conducted, including the analysis of the average effect size, 95% confidence intervals (CI), *Q*_t_, and I^2^ [30]. To assess the extent to which different independent variables are influenced by temperature, meta-analyses were conducted separately to determine the degree of temperature impact on various stages of *S. frugiperda*.

The heterogeneity statistic is a test of the weighted sum of squares against a chi-square distribution with k-1 degrees of freedom. When the 95% confidence interval of the effect size includes 0, it indicates that the effect size of the experimental group is equal to that of the control group, and the impacts of both on the study subject are comparable (*p* > 0.05). When the values of the 95% confidence interval are all greater than 0, it indicates that the effect size of the experimental group is greater than that of the control group (*p* < 0.05). Conversely, when the values of the 95% confidence interval are all less than 0, it indicates that the effect size of the experimental group is less than that of the control group (*p* < 0.05) [31]. Based on the significance of the cumulative effect size with respect to zero and the *p*-value of *Q_t_*, we determined whether explanatory variables need to be included. The explanatory variables to be considered included the influence of relative humidity and photoperiod on the cumulative effect size. Furthermore, temperature data were treated as a continuous variable to determine their impact on the mean effect size. In a meta-analysis, overall heterogeneity is divided into variance explained by categorical factors (between-group heterogeneity) and residual variance (within-group heterogeneity), and their significance can be determined through k-1 testing [32].

To test for potential publication bias, a funnel plot was used to examine the relationship between effect size and sample size. The significance of the *p*-value can indicate whether publication bias exists in the study [33]. If the statistically significant results remain unchanged after correction, it can be concluded that the results are stable and not influenced by publication bias.

## 3. Results

### 3.1. Statistical Data

Our study covers results from 20 publications, comprising a total of 605 observations. The research involves 16 dependent variables related to the temperature variations of *S. frugiperda*, including the following: first instar (*n* = 36), second instar (*n* = 36), third instar (*n* = 36), fourth instar (*n* = 36), fifth instar (*n* = 36), sixth instar (*n* = 36), adult longevity (*n* = 36), eclosion rate (*n* = 26), egg (*n* = 38), egg to adult (*n* = 39), fecundity (*n* = 24), life cycle (*n* = 33), oviposition period (*n* = 39), pre-oviposition period (*n* = 37), pupa developmental duration (*n* = 39), and pupal period (*n* = 52) (Table 1).

### 3.2. Overall Random-Effects and Fixed-Effects Meta-Analyses

The results indicate that temperature elevation enhances the adaptability of *S. frugiperda* across the studies, with a total mean effect size of −0.5668 (CI: −0.6170, −0.5165; Figure 1). Apart from the oviposition period, which increases with temperature elevation, and the adult lifespan, which increases with temperature elevation, with no significant change in the eclosion rate, all other dependent variables related to *S. frugiperda* showed a significant decrease with temperature elevation (Figure 2). Treating temperature as a continuous variable, observations revealed the overall changes in *S. frugiperda* across various temperature gradients. After temperatures rise above 20 °C, all physiological indicators of *S. frugiperda* show a significant enhancement (Figure 3A). When the temperature reaches 32 °C, the physiological activity of *S. frugiperda* peaks (Figure 3B).

### 3.3. Effect of Temperature on Development Period

In every investigation, temperature elevation led to a reduction in the developmental period of *S. frugiperda* eggs, with a total mean effect size of −0.8841 (CI: −1.0609, −0.7073; Appendix A). Within the temperature threshold range of 15–36 °C, the developmental time of *S. frugiperda* eggs significantly decreases with increasing temperature (Figure 4A). The calculation results of both the random-effects model and fixed-effects model show that *Q* (df = 37) = 51,662.5037, *p* < 0.0001, indicating significant between-group differences affecting the cumulative effect size, necessitating the introduction of explanatory variables such as humidity and photoperiod (Figure 4B). When the temperature reaches 30 °C, the relative humidity is 75%, and the photoperiod is 12:12, it is most suitable for the development of *S. frugiperda* eggs.

In each study, temperature elevation leads to a shorter developmental period from egg to adult for *S. frugiperda*, with a total mean effect size of −0.2560 (CI: −0.3857, −0.1263; Appendix A). Within the temperature threshold range of 18–36 °C, the developmental time from egg to adult for *S. frugiperda* significantly decreases with increasing temperature (Figure 4A). The calculation results of both the random-effects model and fixed-effects model show that *Q* (df = 38) = 21,034.5417, *p* < 0.0001, indicating significant between-group heterogeneity affecting the cumulative effect size, necessitating the inclusion of explanatory variables (Figure 4C). The results indicate that humidity (*Q_m_* = 10.2304, *p* = 0.0167) is one of the factors influencing the cumulative effect size, with differences in humidity affecting the developmental time from egg to adult for *S. frugiperda* differently. When the temperature reaches 32 °C with a relative humidity of 65% and a photoperiod of 12:12, these are the most suitable external environmental conditions for the development of *S. frugiperda* eggs into adults.

Across all research, temperature elevation results in a shorter developmental period for the first instar stage of *S. frugiperda*, with a total mean effect size of −0.9155 (CI: −1.0556, −0.7754; Appendix A). Within the temperature threshold range of 18–36 °C, the developmental time of the first instar stage of *S. frugiperda* significantly decreases with increasing temperature (Figure 5A). The calculation results of both the random-effects model and fixed-effects model show that *Q* (df = 35) = 1014.9817, *p* < 0.0001, indicating significant between-group differences affecting the cumulative effect size, necessitating the inclusion of explanatory variables (Figure 5B). The results indicate that humidity (*Q_m_* = 23.4597, *p* < 0.0001) and photoperiod (*Q_m_* = 16.2425, *p* < 0.0001) are factors influencing the cumulative effect size, with differences in the relative humidity and photoperiod affecting the impact of temperature variation on the first instar developmental period of *S. frugiperda* differently. When the temperature reaches 27 °C with a relative humidity of 75% and a photoperiod of 12:12, these are the most suitable conditions for the development of first instar stage of *S. frugiperda*.

In every experiment, temperature elevation leads to a shorter developmental period for the second instar stage of *S. frugiperda*, with a total mean effect size of −0.8096 (CI: −0.9425, −0.6766; Appendix A). Within the temperature threshold range of 18–36 °C, the developmental time of the second instar stage of *S. frugiperda* significantly decreases with increasing temperature (Figure 5A). The calculation results of both the random-effects model and fixed-effects model show that *Q* (df = 35) = 2035.6998, *p* < 0.0001, indicating significant between-group heterogeneity affecting the cumulative effect size, necessitating the inclusion of explanatory variables (Figure 5C). The results indicate that photoperiod (*Q_m_* = 5.2683, *p* = 0.0217) is one of the factors influencing the developmental period of the second instar stage of *S. frugiperda*. When the temperature reaches 32 °C with a relative humidity of 75% and a photoperiod of 12:12, these are the most suitable conditions for the development of second-instar larvae.

Within all studies, temperature elevation resulted in a shorter developmental period for the third instar stage of *S. frugiperda*, with a total mean effect size of −0.8946 (CI: −1.0507, −0.7386; Appendix A). Within the temperature threshold range of 18–36 °C, the developmental time of the third instar stage of *S. frugiperda* significantly decreases with increasing temperature (Figure 5A). The calculation results of both the random-effects model and fixed-effects model show that *Q* (df = 35) = 6709.9338, *p* < 0.0001, indicating significant between-group heterogeneity affecting the cumulative effect size, necessitating the inclusion of explanatory variables (Figure 5D). When the temperature reaches 32 °C with a relative humidity of 75% and a photoperiod of 12:12, these are the most suitable conditions for the development of third-instar larvae.

In all cases, temperature elevation resulted in a shorter developmental period for the fourth instar stage of *S. frugiperda*, with a total mean effect size of −0.8952 (CI: −1.0476, −0.7427; Appendix A). Within the temperature threshold range of 18–36 °C, the developmental time of the fourth instar stage of *S. frugiperda* significantly decreases with increasing temperature (Figure 5E). The calculation results of both the random-effects model and fixed-effects model show that *Q* (df = 35) = 13,509.9649, *p* < 0.0001, indicating significant between-group heterogeneity affecting the cumulative effect size, necessitating the inclusion of explanatory variables (Figure 5F). When the temperature reaches 30 °C with a relative humidity of 75% and a photoperiod of 12:12, these are the most suitable conditions for the development of fourth-instar larvae.

For every study, temperature elevation resulted in a shorter developmental period for the fifth instar stage of *S. frugiperda*, with a total mean effect size of −0.9214 (CI: −1.0697, −0.7732; Appendix A). Within the temperature threshold range of 18–36 °C, the developmental time of the fifth instar stage of *S. frugiperda* significantly decreases with increasing temperature (Figure 5E). The calculation results of both the random-effects model and fixed-effects model show that *Q* (df = 35) = 1014.9817, *p* < 0.0001, indicating significant between-group differences affecting the cumulative effect size, necessitating the inclusion of explanatory variables (Figure 5G). The results indicate that humidity (*Q_m_* = 7.1220, *p* = 0.0284) and photoperiod (*Q_m_* = 7.0925, *p* = 0.0077) are factors influencing the cumulative effect size, with differences in the relative humidity and photoperiod affecting the impact of temperature variation on the fifth instar developmental period of *S. frugiperda* differently. When the temperature reaches 30 °C with a relative humidity of 75% and a photoperiod of 12:12, these are the most suitable conditions for the development of fifth-instar larvae.

Within each experiment, temperature elevation led to a shorter developmental period for the sixth instar stage of *S. frugiperda*, with a total mean effect size of −1.0217 (CI: −1.1758, −0.5677; Appendix A). Within the temperature threshold range of 18–36 °C, the developmental time of the fourth instar stage of *S. Frugiperda* significantly decreases with increasing temperature (Figure 5E). The calculation results of both the random-effects model and fixed-effects model show that *Q* (df = 35) = 5289.4684, *p* < 0.0001, indicating significant between-group heterogeneity affecting the cumulative effect size, necessitating the inclusion of explanatory variables (Figure 5H). When the temperature reaches 30 °C with a relative humidity of 75% and a photoperiod of 12:12, these are the most suitable conditions for the development of sixth-instar larvae.

### 3.4. Effect of Temperature on Female Oviposition Behavior

For each analysis, an increase in temperature led to an increase in the oviposition quantity of *S. frugiperda* females, with a total average effect size of −0.2190 (CI: −0.2245, −0.2136; Appendix A). Within the temperature range of 10–41 °C, there was a significant increase in the reproductive capacity of *S. frugiperda* with rising temperatures (Figure 6A). The results from the random-effects model and fixed-effects model calculations indicated that *Q* (df = 38) = 21,034.5417, *p* < 0.0001, indicating the presence of between-group heterogeneity affecting the cumulative effect size, necessitating the inclusion of explanatory variables (Figure 6B). When the temperature is 19 °C with a relative humidity of 70% and a photoperiod of 16:8, the female moth lays the highest number of eggs.

Within all our studies, an increase in temperature led to an extension of the oviposition period for *S. frugiperda* females, with a total average effect size of 0.6072 (CI: 0.4465, 0.7678; Appendix A). Within the temperature range of 10–41 °C, there was a significant increase in the oviposition period of *S. frugiperda* with rising temperatures (Figure 6A). The results from the random-effects model and fixed-effects model calculations indicated that *Q* (df = 38) = 94.5784, *p* < 0.0001, indicating the presence of between-group heterogeneity affecting the cumulative effect size, necessitating the inclusion of explanatory variables (Figure 6C). The findings revealed that relative humidity (*Q_m_* = 11.6077, *p* = 0.003) and photoperiod (*Q_m_* = 35.8915, *p* < 0.0001) were, respectively, among the factors affecting the cumulative effect size, with different effects of the photoperiod and humidity on the oviposition period of *S. frugiperda*. When the temperature is 18 °C with a relative humidity of 75% and a photoperiod of 15:9, the adult female moth has the longest oviposition period.

In each study, an increase in temperature resulted in a decrease in the pre-oviposition period for *S. frugiperda* females, with a total average effect size of −0.4103 (CI: −0.5345, −0.2861; Appendix A). Within the temperature range of 10–41 °C, there was a significant decrease in the pre-oviposition period of *S. frugiperda* with rising temperatures (Figure 6A). The results from the random-effects model and fixed-effects model calculations showed that *Q* (df = 36) = 298.3982, *p* < 0.0001, indicating the presence of between-group heterogeneity affecting the cumulative effect size, necessitating the inclusion of explanatory variables (Figure 6D). The findings indicated that photoperiod (*Q_m_* = 14.2962, *p* = 0.0025) and relative humidity (*Q_m_* = 11.8621, *p* = 0.0027) were, respectively, among the factors influencing the cumulative effect size, with different effects of the photoperiod and humidity on the pre-oviposition period of *S. frugiperda*. When the temperature is 25 °C with a relative humidity of 70% and a photoperiod of 14:10, the pre-oviposition period of adult females is the shortest.

### 3.5. The Impact of Temperature Variation on Pupal Development of S. frugiperda

In various studies, an increase in temperature has been found to result in a shortened pupal developmental period of *S. frugiperda*, with a total average effect size of −0.6780 (CI: −0.8597, −0.4963; Appendix A). Within the temperature range of 10–38 °C, there is a significant decrease in the pupal developmental period of *S. frugiperda* with increasing temperature (Figure 7A). The results from both the random-effects and fixed-effects models indicate that *Q* (df = 38) = 5646.9218, *p* < 0.0001, suggesting the presence of between-group heterogeneity influencing the cumulative effect size, necessitating the inclusion of explanatory variables (Figure 7B). The findings suggest that the photoperiod (*Q_m_* = 74.8477, *p* < 0.0001) is one of the factors influencing the cumulative effect size, with different photoperiods having varying effects on the pupal developmental period of *S. frugiperda*. When the temperature reaches 30 °C and a photoperiod of 14:10, it is the most suitable developmental period for pupae.

In various studies, an increase in temperature has been observed to shorten the pupal stage of *S. frugiperda*, with a total average effect size of −0.7901 (CI: −0.8082, −0.7719; Appendix A). Within the temperature threshold range of 18–36 °C, the duration of the pupal stage of *S. frugiperda* significantly decreases with increasing temperatures (Figure 7A). The results from both the random-effects and fixed-effects models indicate that Q (df = 38) = 5646.9218, *p* < 0.0001, indicating between-group differences influencing the cumulative effect size, necessitating the inclusion of explanatory variables (Figure 7C). The findings suggest that relative humidity (*Q_m_* = 9.4457, *p* = 0.0239) is one of the factors influencing the cumulative effect size, with different humidities having varying effects on the pupal stage of *S. frugiperda*. When the temperature reaches 30 °C, the relative humidity is 75%, and the light–dark (L) ratio is 12:12, the developmental period of the pupal stage is shortest.

In all cases, the effect of temperature variation on the Eclosion rate of *S. frugiperda*. was found to be insignificant, with a total average effect size of 0.0134 (CI: −0.1098, 0.1365; Appendix A). Due to the results computed by the random-effects model, the confidence interval intersects with 0, indicating that within the temperature threshold range of 10–38 °C, the impact of temperature increase on the Eclosion rate of *S. frugiperda* is not significant.

### 3.6. The Impact of Temperature Variation on the Lifespan of S. frugiperda

In all cases, an increase in temperature has been found to lead to an extension of the lifespan of adult *S. frugiperda*, with a total average effect size of −0.1016 (CI: −0.2252, −0.0220; Appendix A). Within the temperature threshold range of 10–41 °C, the lifespan of adult *S. frugiperda* significantly increases with rising temperatures (Figure 8A). The results from both the random-effects and fixed-effects models reveal that *Q* (df = 61) = 3246.3144, *p* < 0.0001, indicating the presence of between-group heterogeneity affecting the cumulative effect size, necessitating the inclusion of explanatory variables (Figure 8B). It is evident that photoperiod (*Q_m_* = 11.5416, *p* = 0.0031) and relative humidity (*Q_m_* = 10.5439, *p* = 0.0012) are among the factors influencing the cumulative effect size, with differing impacts on the lifespan of adult *S. frugiperda*. When the temperature reaches 32 °C, the relative humidity is 75%, and the light–dark (L) ratio is 12:12, the lifespan of adult *S. frugiperda* is shortest.

For every study, an increase in temperature was found to result in a shortened life cycle of *S. frugiperda*, with a total average effect size of −0.5349 (CI: −0.6366, −0.4331; Appendix A). Within the temperature threshold range of 20–36 °C, the life cycle of *S. frugiperda* significantly decreases with rising temperatures (Figure 8A). Results from both the random-effects and fixed-effects models indicate that *Q* = (df = 32) = 12,303.1473, *p* < 0.0001, suggesting the presence of between-group heterogeneity affecting the cumulative effect size, necessitating the inclusion of explanatory variables (Figure 8C). It is evident that the photoperiod (*Q_m_* = 12.7604, *p* = 0.0004) and relative humidity (*Q_m_* = 12.7604, *p* = 0.0004) are among the factors influencing the cumulative effect size, with differing impacts on the life cycle of *S. frugiperda*. When the temperature reaches 36 °C with a relative humidity of 75% and a light–dark (L) ratio of 12:12, the life cycle of *S. frugiperda* is shortest.

### 3.7. Model Testing

Funnel plots and radar charts were used to assess whether the results were influenced by publication bias, and the reliability of our results was validated through the calculation of the fail-safe coefficient. The findings indicate that the reliability of our results is confirmed by the funnel plot (Figure 9A) (*z* = 11.7603, *p* = 0.0702), radar chart (Figure 9B), and fail-safe coefficient.

## 4. Discussion

Under the conditions of global warming, the distribution range of *S. frugiperda* is expected to significantly increase [34,35]. In a warmer world, does the adaptability of *S. frugiperda* increase? To answer this question, we conducted a meta-analysis of 20 relevant reports by systematically retrieving and screening them and calculated the changes in *S. frugiperda*’s temperature response. The results indicate that within the temperature threshold range of 10–41 °C, the adaptability of *S. frugiperda* gradually strengthens with increasing temperature. When the temperature reaches 32 °C, *S. frugiperda* has the strongest adaptability, which is consistent with previous research findings [36,37,38,39,40,41,42,43,44,45]. This further confirms the reliability of the model. Typically, the longer the lifespan of an insect, the longer it has for reproduction, allowing more time for egg-laying. If the lifespan of adult insects is shortened at 32 °C, the time available for reproduction will also be reduced. Therefore, a shorter lifespan generally leads to a decrease in the number of eggs laid, thereby reducing fecundity.

Given the potential for the significant damage and economic losses that *S. frugiperda* may cause in agricultural planting areas, monitoring its adaptability is particularly important [46]. High temperatures typically accelerate the development of *S. frugiperda*. The development rate of insects is closely related to temperature, and higher temperatures can shorten the duration of the egg, larval, pupal, and adult stages. Therefore, under high temperature conditions, the entire life cycle is shortened. This study, through meta-analysis, predicts that the adaptability of *S. frugiperda* reaches its peak at the temperature of 32 °C. Given the importance of early warning for insect invasion directions and suitable ranges, this research will provide crucial information for future monitoring and alerts (Table 2).

As is well-known, insects are poikilothermic animals, and the distributional changes of insect populations are strongly influenced by temperature [47,48]. Therefore, the distribution patterns of many insect species are significantly affected by ongoing global warming, leading to range expansions and alterations [49,50]. In this scenario, temperature could be the primary driver for the survival, development, and reproduction of *S. frugiperda*. Temperature significantly influences the growth and development cycle of *S. frugiperda*, with growth being promoted within the suitable temperature range. We also analyzed the relationship between the different stages of *S. frugiperda* and temperature, obtaining corresponding response curves. We observed that each developmental stage of *S. frugiperda* varies with temperature changes.

Continued global warming will greatly alter the functionality and structure of ecosystems, leading to changes in the distribution of biological habitats [51]. The increasing incidence and expanding distribution of insect infestations and their associated damages will have profound implications for agricultural production [52]. Under future climate warming conditions, it is expected that the adaptability of *S. frugiperda* will gradually strengthen. This study’s findings offer policymakers insights to formulate more effective pest management strategies, aiming to mitigate the potential widespread economic losses caused by pests under future climate warming.

## 5. Conclusions

The fall armyworm is an important agricultural pest of various crops; therefore, a reliable method is needed to predict its occurrence in the field. Temperature-dependent development and thermal biology parameters are commonly used methods for simulating insect phenology. In this study, we evaluated the adaptability of *S. frugiperda* under different temperatures. In this study, 602 warming experiment datasets were synthesized, and meta-analysis was employed to predict the suitable temperature range for the survival of *S. frugiperda*. The results indicate that within the temperature range of 10–41 °C, the adaptability of *S. frugiperda* increases with rising temperature. When the temperature reaches the range of 30–35 °C, it is most conducive to the growth of *S. frugiperda*. Over the past few decades, the distribution range of *S. frugiperda* has continued to expand, but more effective management methods are currently lacking. It is crucial to note that timing is essential for chemical control. Therefore, early prediction of suitable temperature ranges and preparation of chemical supplies will achieve better preventive effects.

## Figures and Tables

**Figure 1 insects-15-00689-f001:**
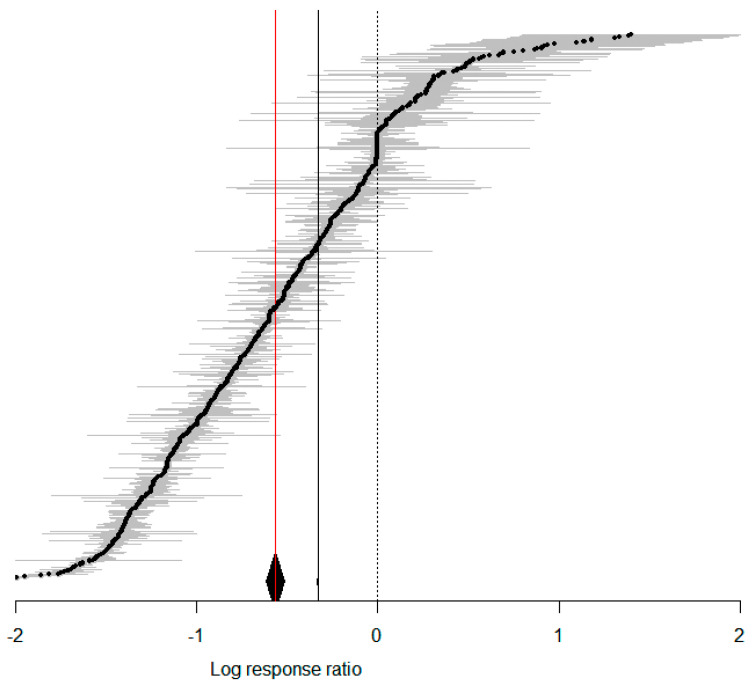
Forest plot of the effects of temperature variation on *S. frugiperda*. (The red line represents the calculated results of the random-effects model, with a cumulative effect size of −0.5668 and 95% confidence interval ranging from −0.6170 to −0.5165. The solid black line represents the calculated results of the fixed-effects model, with a cumulative effect size of −0.3320 and 95% confidence interval ranging from −0.3327 to −0.3312. The black dashed line represents x = 0. Gray lines represent the standard error of a single factor).

**Figure 2 insects-15-00689-f002:**
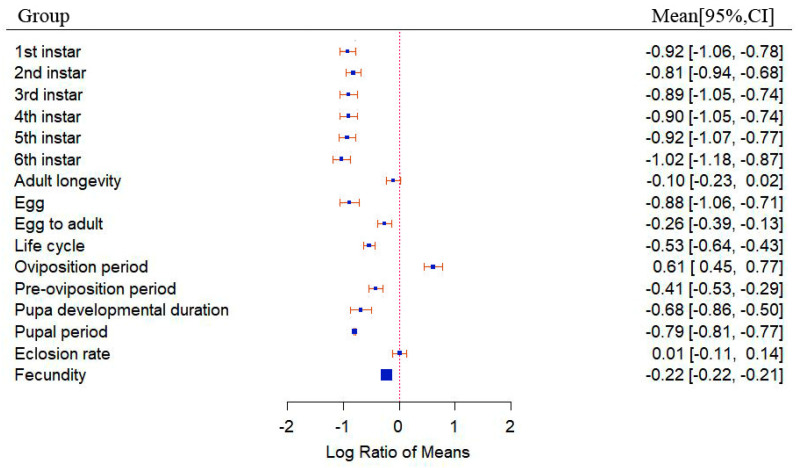
Forest plot of the effects of temperature variation on various physiological indicators of *S. frugiperda*. (The blue squares represent the values of the cumulative effect size for each group. The red dashed line represents x = 0. The orange lines represent the 95% confidence interval).

**Figure 3 insects-15-00689-f003:**
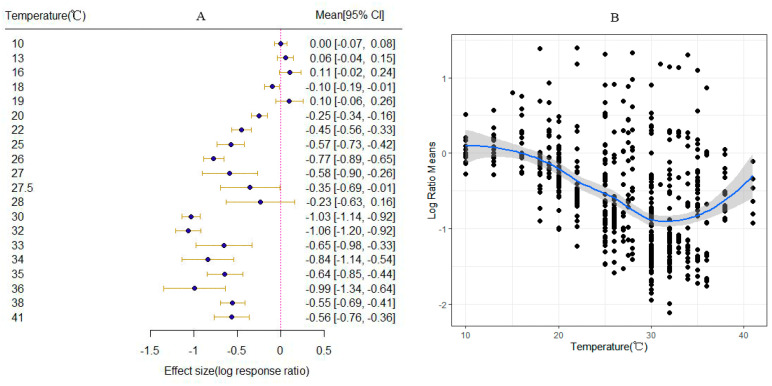
The impact of temperature variation on *S. frugiperda*. ((**A**) depicts the changes in physiological properties of *S. frugiperda* with increasing temperature. The red dashed line represents x = 0. The blue squares represent the values of the cumulative effect size for each temperature gradient. The yellow lines represent the 95% confidence interval; (**B**) shows the temperature range curve for the optimal growth of *S. frugiperda*. Black solid dots represent all factors. The blue solid line indicates the suitability of the fall armyworm to temperature changes. The shaded area represents the 95% confidence interval).

**Figure 4 insects-15-00689-f004:**
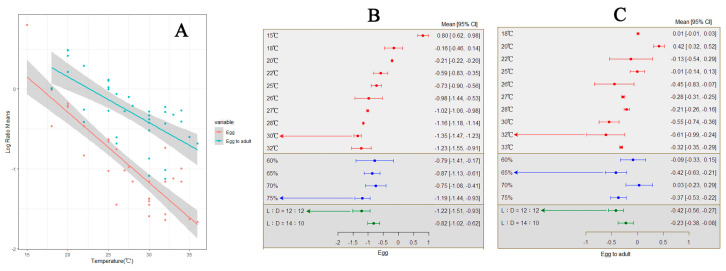
The effect of temperature on the incubation and development time of *S. frugiperda* eggs into adult insects. ((**A**) shows that as the temperature increases, the incubation time of the eggs gradually shortens; (**B**,**C**) display the optimal external environmental conditions for egg incubation and the development of eggs into adults).

**Figure 5 insects-15-00689-f005:**
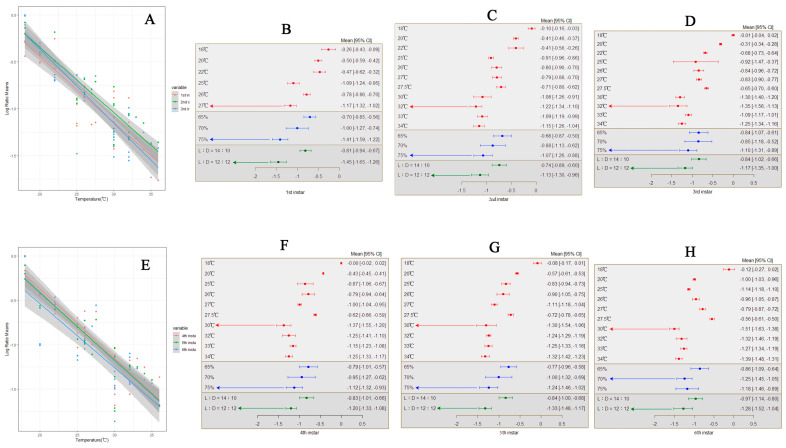
The impact of temperature on various developmental stages of *S. frugiperda.* ((**A**) illustrates the changes in developmental time of first-to-third-instar larvae with temperature variations; (**B**–**D**) depict the response of first-to-third-instar larvae to external environmental conditions; (**E**) illustrates the changes in developmental time of fourth-to-sixth-instar larvae with temperature variations; (**E**–**H**) depict the response of fourth-to-sixth-instar larvae to external environmental conditions).

**Figure 6 insects-15-00689-f006:**
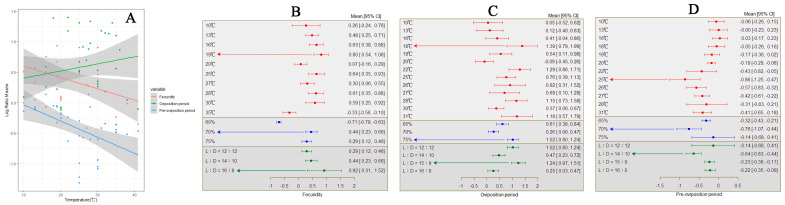
The impact of temperature on the oviposition behavior of *S. frugiperda.* ((**A**) illustrates the variation in oviposition behavior of female fall armyworms with temperature changes; (**B**) depicts the response of oviposition quantity of the female *S. frugiperda* to changes in external environmental conditions; (**C**) shows the response of oviposition period of the female *S. frugiperda* to changes in external environmental conditions; (**D**) demonstrates the response of pre-oviposition period of the female *S. frugiperda* to changes in external environmental conditions).

**Figure 7 insects-15-00689-f007:**
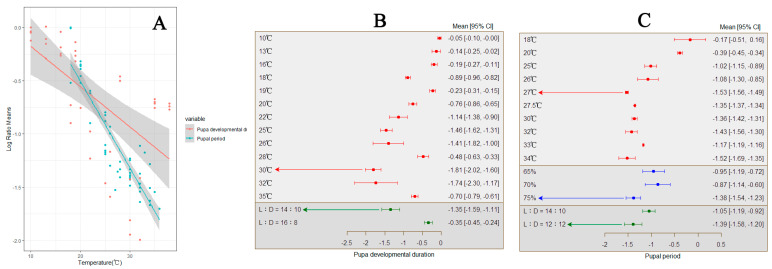
The influence of temperature variation on the development of *S. frugiperda* pupae. ((**A**) represents the change in the developmental period of *S. frugiperda* pupae with increasing temperature; (**B**) represents the response of *S. frugiperda* pupae’s developmental period to changes in external environmental conditions; (**C**) represents the response of the entire pupal period of armyworms to changes in external environmental conditions).

**Figure 8 insects-15-00689-f008:**
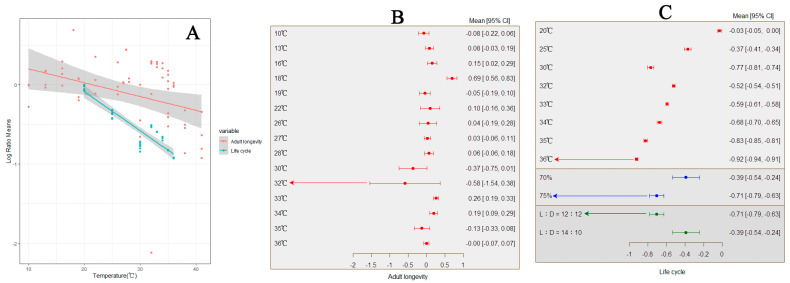
The influence of temperature variation on the lifespan of *S. frugiperda*. ((**A**) represents the change in the lifespan and life cycle of adults with increasing temperature; (**B**) represents the response of adult lifespan to changes in external environmental conditions; (**C**) represents the response of the life cycle of to changes in external environmental conditions).

**Figure 9 insects-15-00689-f009:**
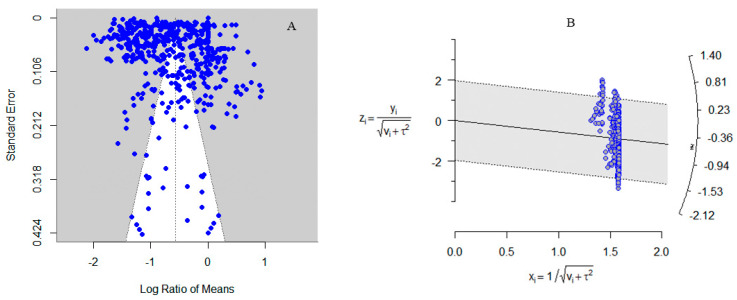
(**A**) is a funnel chart; (**B**) is a radar chart.

**Table 1 insects-15-00689-t001:** Dataset on physiological indicators of *S. frugiperda*.

Temperature Tange	Control Temperature	Number of Data Sets (*n*)	Variable
18–36 °C	18 °C	36	1st instar
18–36 °C	18 °C	36	2nd instar
18–36 °C	18 °C	36	3rd instar
18–36 °C	18 °C	36	4th instar
18–36 °C	18 °C	36	5th instar
18–36 °C	18 °C	36	6th instar
10–41 °C	10 °C	62	Adult longevity
10–38 °C	10 °C	26	Eclosion rate
15–36 °C	15 °C	38	Egg
18–36 °C	18 °C	39	Egg to adult
10–41 °C	10 °C	24	Fecundity
20–36 °C	20 °C	33	Life cycle
10–41 °C	10 °C	39	Oviposition period
10–41 °C	10 °C	37	Pre-oviposition period
10–38 °C	10 °C	39	Pupa developmental duration
18–36 °C	18 °C	52	Pupal period
10–41 °C	10 °C	605	*S. frugiperda*

**Table 2 insects-15-00689-t002:** The optimal external environment for each developmental stage of *S. frugiperda*.

Growth History	Optimal Survival Temperature	Optimum RH for Survival	Optimum Photoperiod for Survival
1st instar	27 °C	75%	L:D = 12:12
2nd instar	32 °C	75%	L:D = 12:12
3rd instar	32 °C	75%	L:D = 12:12
4th instar	30 °C	75%	L:D = 12:12
5th instar	30 °C	75%	L:D = 12:12
6th instar	30 °C	70%	L:D = 12:12
Adult longevity	32 °C	65%	L:D = 14:10
Egg	30 °C	75%	L:D = 12:12
Egg to adult	32 °C	65%	L:D = 12:12
Life cycle	36 °C	75%	L:D = 12:12
Oviposition period	18 °C	75%	L:D = 15:9
Pre-oviposition period	25 °C	70%	L:D = 14:10
Pupa developmental Duration	30 °C	-	L:D = 14:10
Pupal period	27 °C	75%	L:D = 12:12
Fecundity	19 °C	70%	L:D = 16:8
Eclosion rate	-	-	-

## Data Availability

The data supporting the results are available in a public repository at: https://doi.org/10.6084/m9.figshare.26968306, accessed on 25 March 2024.

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
