# Peer review of "Effects of Global Climate Warming on the Biological Characteristics of Spodoptera frugiperda (J.E. Smith) (Lepidoptera: Noctuidae)"

_insects, 2024, doi:10.3390/insects15090689_

Round 1
Reviewer 1 Report
Comments and Suggestions for Authors
This manuscript describes a thorough review of the literature regarding climatic variables (such as temperature, relative humidity, light-and-dark cycle, etc.) on the various life stages of the fall armyworm, Spodoptera frugiperda or S. frugiperda) using a meta-analysis. Note to authors: through the manuscript, it is sufficient to simply say “S. frugiperda” not “The S. frugiperda”. That said, there are other editorial issues which the authors should address, noted below.
Methodologically, the science behind the paper seems sound, however there are several issues noted here:
-beginning on line 219, in the results section, each paragraph begins with “In each study…”; as multiple paragraphs begin with this, it could make the reading easier and less tedious to express this differently; however, this is just a minor editorial suggesting on my part that is not completely necessary for the authors to change.
-Figures 5 and 6; the text within the figures is very hard to read in my review copy.
-line 413, if the lifespan of adult S. frugiperda is shortest at 32 °C, how does this affect its fecundity? Would it not decrease it?
-also line 416, how does the life cycle of S. frugiperda decrease?
-line 435, where and how is the fail-safe coefficient calculated?
-The titles above the two items in Figure 9 should be changed.
-line 482, in the temperature range of 10 – 41 °C, doesn’t performance of the insect drop off near the top of this range (ie. above 32 °C)? So maybe the adaptability doesn’t keep increasing with temperatures up to 41 °C?
Editorial comments:
-maybe the common name of S. frugiperda (fall armyworm) could be mentioned in the abstract and introduction.
-line 19, there should be a space after the period, before “However”. This editorial error occurs elsewhere throughout the manuscript.
-line 24, “it” should be capitalized. This occurs elsewhere in the manuscript.
-line 25, omit “to”
-line 27, I recommend “highest point” instead of “most active”
-line 28, omit “reduced”.
-line 29, I recommend “leads” not “leading”.
-line 34, I recommend adding “S. frugiperda” to the keywords.
-line 50, I recommend “…crop pest in the Americas. Its larvae can attack…” (use 2 sentences in stead of 1).
-line 91, I recommend “included” not “inclusion”.
-line 105, I recommend “The changes in various physiological indicators were investigated…” as an alternative wording.
-line 107, I recommend “data were extracted.” not “extract data”.
-line 109, maybe “contains standard error” not “is standard error” as an alternative wording.
-line 116, maybe “The log response ratios was chosen…” not “…is chosen…” (different verb tense). This occurs elsewhere in the manuscript; there should be consistency throughout the manuscript regarding the tense of the verb.
-line 125, maybe mention Formula 4 in the paragraph before formula 4, not in the paragraph before formula 3.
-line 136, the standard error is s(RR+), not just (RR+), correct?
-line 140, Qt not Qt
-line 171, “the” not “he”, also omit “factors”.
-line 179, maybe “The significance of the p-value can indicate…” not “The significance p-value…” as an alternative wording.
-line 188, “including” not “include”
-line 203, “between 32 °C” is not a range; from the wording of the sentence, a temperature range is expected. Likewise on line 450.
-line 357, “females” not “female”
-line 385 “relative humidity” not “relatively humidity”
-line 449, omit “In” at he beginning of the sentence.
-lines 454 and 455 should be removed.
-references 12 and 25, is there really a need to capitalize the authors’ names?
Comments on the Quality of English LanguageNA
Reviewer 2 Report
Comments and Suggestions for Authors
The authors finding regarding ‘one of the most destructive crop pests-hosts numbering in excess of 300’ has relevance world-wide. The invasive success of Spodoptera frugiperda is well known and guidance as to future prediction for range expansion and control potential are clearly important.
The authors use a thorough literature search and subsequent metanalysis using selected studies to confirm that the problem is going to get worse under predicted climate changes. As such the information here can be used to increase effectiveness of monitoring.
The thorough literature search and identification of studies which fit screening criteria is excellent, followed by robust statistical analysis, gives confidence in the conclusion drawn from the analysis.
Include photoperiod in analysis-include some mention of role of photoperiod in life cycle in the introduction would be interesting.
L46-47 the influence of the lack of coevolved host plants may be limited in a highly polyphagous crop pest such as S. frugiperda
his study, through Meta-analysis, predicts that the adaptability of the S. frugiperda reaches its peak within the temperature threshold range of 32℃. Given the importance of early warning for insect invasion directions and suitable ranges, this research will provide crucial information for future monitoring
Further, the importance for environmental impacts of optimizing chemical use for pest control with minimal environment al impacts and as timing is essential for chemical control information regarding response to anticipated changes in timing of occurrence and distribution will contribute to control
L 359, 365 and elsewhere no capital on frigiperda
L 450 edits needed
Round 2
Reviewer 1 Report
Comments and Suggestions for Authors
Thank you for making the suggested corrections. You have answered all of my questions and concerns, however, there are just several small editorial issues remaining:
-line 103, "By observing changes in various physiological indicators." This is a sentence fragment; I am not sure how it ties in to the sentence before and after; I don't want to suggest a rewording because this might obscure the authors' meaning. This should be easy to fix.
-line 128, equation 3, the variance (v) is calculated in equation 3, correct? The authors state it is calculated in equation 4, however (line 127). Unless I misunderstand, shouldn't the sentence on lines 126-7 read "The variance (v) (formula (3)) of the RR is calculated as follows (formula (3)):"
-line 127, there should be a space after I2. This also occurs on line 271
-line 180-182, this sentence is repeated; please omit one of the two.
-Figure 3, Panel B should have "B" label over it.
-line 444, I suggest "In a warmer world,..." not "A warmer world..."
-just "S. frugiperda", not "the S. frugiperda", lines 444, 447, 448, 456, 462, 497, 498, 500, 512, 515, 521.
-line 461, I suggest "This study, through meta-analysis..." not "In this study, through meta-analysis..."
-Table 2 cuts off some of the text of the discussion; please correct this.
-line 463, 32 degrees C is not a range.
Comments on the Quality of English Language
No concerns at this point, just some minor typos / editorial issues noted in my comments to the authors.
